# New Aspects of In Situ Measurements for Downy Mildew Forecasting

**DOI:** 10.3390/plants11141807

**Published:** 2022-07-08

**Authors:** Melissa Kleb, Nikolaus Merkt, Christian Zörb

**Affiliations:** Institute of Crop Science, Department of Quality of Plant Products, Viticulture (340e), University of Hohenheim, Emil-Wolff-Strasse 35, 70599 Stuttgart, Germany; nikolaus.merkt@uni-hohenheim.de (N.M.); christian.zoerb@uni-hohenheim.de (C.Z.)

**Keywords:** downy mildew, forecasting, grapevine canopy, VitiMeteo

## Abstract

Downy mildew is, globally, one of the most significant diseases in viticulture. Control of this pathogen is achieved through fungicide application. However, due to restrictions (from upcoming regulations) and growing environmental conscientiousness, it is critical to continuously enhance forecasting models to reduce fungicide application. Infection potential has traditionally been based on a 50 h–degree calculation (temperature multiplied by leaf wetness duration) measured by weather stations; the main climatic parameters for forecast modelling are temperature, relative humidity, and leaf wetness. This study took these parameters measured by a weather station and compared them with the same parameters measured inside a grape canopy. The study showed that the temperature readings by the weather station compared to inside the canopy recorded differences during the day but not at night; the relative humidity showed significant differences during both daytime and night; leaf wetness showed the highest differences and was statistically significant during both daytime and night. In conclusion, the measurement differences between inside of the canopy and at the weather station have significant impacts on the precision of forecasting models. Thus, using data from inside of a canopy for the prediction should lead to even less fungicide applications.

## 1. Introduction

Recent extreme weather events and atypical seasons are indications of climatic change. For example, the summer of 2021 in Germany was very humid compared to past years. The variations from historic seasonal weather patterns may necessitate changes in plant cultivation and may further necessitate an adaptation of plant protective treatments. An extremely humid climate causes notable problems of fungal pathogens, especially the increased infection risk of downy mildew (Plasmopara viticola). Downy mildew is a serious problem in viticulture, causing plant and berry damage, which may lead to quality- and harvest losses. The spreading and multiplying of the pathogen require available water at the abaxial leaf surface, enabling released zoospores to “swim” to reach stomata for entrance and infection. Just as important as leaf wetness are temperature and relative humidity. Both strongly influence the life span of the sporangia. Higher temperatures require higher relative humidity (and lower temperatures require lower relative humidity) for the survival of the sporangia. At temperatures above 30 °C for more than 6 h, sporangia die regardless of the relative humidity value. To easily calculate the probability of infection in general, the 50 h–degree rule of thumb can be applied (hour-degree = temperature[°C] * leaf wetness duration[h]) [1,2]. However, downy mildew not only appears in extremely humid years—the pathogen appears more or less every year and in every vineyard. A considerable number of fungal-resistant grapevine varieties have been cultivated. These varieties may persist through infective periods without the need for synthetic fungal plant protection. Over four decades of research shows that the wine quality and taste of new resistant varieties have increased compared to the early days of breeding. However, consumers, in general, are less open to new grape varieties and are unaware of the viticultural benefits of new resistant varieties (and are, therefore, sceptical). Therefore, the market share of new resistant varieties is less compared to non-resistant conventional varieties [3]. In conventional varieties, the only way to keep pathogen spread under control is by using fungicides. The application of copper or synthetic pesticides is costly, time-consuming, and may negatively affect the environment. Moreover, increasing environmental conscientiousness and regulations, especially by the EU [4,5], promotes a continuous reduction of these fungicide applications. Forecasting models—first developed in Germany, France, Switzerland, Italy, and Australia [6]—therefore help to monitor and predict the favourable conditions for the pathogens by using climatic data from field weather stations. This prediction method can help to reduce many unnecessary preventive applications. One of the most sophisticated prediction models is VitiMeteo, which was invented in 2002–2003 [7]. The precursors of this computer-based prediction model started in 1996 with “BIOMAT” and “HP 100” [8]. Today, VitiMeteo is well established in Germany and Switzerland and has a great influence on ecologically-managed vineyards. VitiMeteo uses data collected from field weather stations and aggregates this information in the AgroMeteo database. Based on the weather station data, the model calculates and predicts the potential infection risk with sporulation and incubation times for specific viticultural regions [9]. Several studies [6,10,11] showed that the prediction of the model is relevant to practice and is highly correlated to real occurrences of infection with accuracy increasing annually. Nevertheless, the precisions of the models have some variations to be minimised. For one, weather stations that are used to provide the prediction for a specific region are sometimes geographically widespread (e.g., at varying distances from vineyards). Additionally, the differences between climatic conditions at a specific vineyard may vary, especially at steep vineyards close to bodies of water. Pfisterer et al. (2021) [12] showed that at steep vineyards, temperature variations from top to bottom were extreme. Additionally, the climate inside a canopy may vary compared to the outside conditions, depending on the canopy’s shape. A weather station may not factor in the impact of the vine–surrounding foliage, which greatly influences the in situ micro-climatic conditions within a grapevine. Therefore, the hypothesis is that weather stations do not convey realistic conditions inside a canopy (or at least can be improved). This discrepancy can lead to variable forecasting accuracies. Peña et al. [13] showed that temperatures inside canopies (at heights of 2 m) and berry clusters showed differences when compared to values respective of weather stations. This study provides evidence for the adjustment of the accuracy of the prediction model parameters to improve fungal disease management at vineyards. To achieve precision, temperature, relative humidity, and leaf wetness were measured at three different heights to account for possible climatic differences from the top to the bottom of the canopy. It is important to note that the results of this study may be applicable as a calculation basis for a prediction model; a feasibility study is in progress.

## 2. Materials and Methods

### 2.1. Location and Plant Material

The experiment site was at the vineyard of the University of Hohenheim in Southern Germany. The examined grapevines were 34-year-old Vitis vinifera L. cv. Pinot meunier grafted on SO4 with a field size of 18 rows and 25 plants per row in a north–south direction. Typical for the region, the vines were trained via a common guyot system with a canopy height of 120 cm. The distance between rows was 180 cm and the distance within the rows was 140 cm.

### 2.2. Experiment Design

The experiment site was divided into three plots (I, II, and III). The division was made to guarantee pathogen-free conditions for each successive measurement period while leaving the examined plot fungicide-free. A highly infected canopy evinces rolled leaves, necrotic leaves, or a loss of leaves, based on the infection intensity. These infection damages would cause different micro-climatic conditions inside the canopy compared to healthy leaves. Each plot had a size of 4 rows with 25 grapevines per row. In every plot, three randomly chosen grapevines (replicates) were equipped with sensors at heights of 100, 130, and 160 cm. Each height had two sensors attached, each sensor including one temperature (T), a relative humidity (rH) gauge, and one leaf wetness (LW) gauge. The sensors were placed inside the canopy as shown in Figure 1a. A standard field weather station (ws) was located at the vineyard 75 m distant from the experiment site. The instruments of the ws were at a height of 180 cm. Data were collected every 10 min. Measured rainfall was used to indicate experiment extremes.

### 2.3. Measurement Periods

The plots were measured from I to III using the same sensors. The measurement period of plot I was from 21 July to 9 August The measurement period of plot II was from 10 August to 31 August and the measurement period of plot III was from 1 September to 27 September (Figure 1b). The periods were divided based on the vegetation statuses of the plants and the status of downy mildew infection of the concerning plot. Each plot was measured from a healthy status up to approximately 25% visible infection. The infection status was captured by visual monitoring. Fungicides against powdery mildew and downy mildew were used in order to ensure pathogen-free conditions when changing the sensors from one plot to the next. Fungicides against powdery mildew were used for all plots during the experiment. Fungicides against downy mildew were used on all but the plot that was under observation. Fungicide applications and substances with concentrations are shown in (Table 1).

### 2.4. Sensors and Sensor Placement

A combined sensor (Testo 174H, Titisee-Neustadt, Germany) was used to measure the air temperature and relative humidity. The sensor output for the temperature was °C with an accuracy of +/− 0.5 °C and a measurement range of −20 °C to +70 °C. The relative humidity is given in rH % with an accuracy of +/− 2% rH. Both T and rH were measured every 10 min. Data were manually collected from data loggers. Data evaluation was conducted using Testo Comfort Software Basic 5.0. For each plot, one Testo 174H sensor was placed at canopy heights of 100, 130, and 160 cm. The leaf wetness was measured by the PHYTOS 31 leaf wetness sensor (METER Group, USA), a sensor comparable to those used by VitiMeteo/AgroMeteo. The PHYTOS 31 sensor measured the leaf surface wetness by measuring the dielectric constant of the sensor’s upper surface. For example, a dry PHYTOS 31 outputs approximately 435 raw counts. The critical threshold was approximately 460 raw counts, which indicates sufficient water for infection [14]. The measurement interval was 10 min. Data were collected via two ZENTRA data loggers (METER Group, USA). Data evaluation was conducted using the ZENTRA Cloud database and Excel. A PHYTOS 31 leaf wetness sensor was placed next to a Testo174H sensor inside the canopy. The leaf wetness sensors were attached at a 45° downward angle and in a south–west direction. The leaf wetness measurements via ws only took place during plot III.

### 2.5. Statistics

Data (T, rH, and LW) were collected every 10 min from inside the canopy and by the ws. For statistical calculation, hourly data were used. Measurements were individually split every day (by sunrise and sunset) for precise representation of daytime and night results. Values were tested for normal distribution and variance homogeneity and were transformed, if necessary. Values of the different canopy heights (n = 4) were analysed using PROC GLM (9.4 (TS1M6), SAS Institute, Inc., Cary, NC, USA). Differences between means were calculated according to Tukey’s test with 5% probability of error. Measured values of the canopy average and weather station were conducted with a pairwise mean comparison of the PROC TTEST. The boxplots were conducted with Excel (version 2016).

## 3. Results

### 3.1. Aggregate Presentation of the Canopy and Weather Station Data

Following figures show the temperature, relative humidity, and leaf wetness for each plot (I, II, and III), collected at three different heights inside the canopy (100, 130, and 160 cm); the values were averaged to represent the whole canopy. Corresponding values were shown as measured by the weather station.

Temperature fluctuation, in general, was caused by seasonal weather fluctuations (Figure 2). During the daytime, measurements taken by the weather station showed constantly lower temperatures compared to inside the canopy. Weather station data never exceeded 30 °C, whereas, inside the canopy, three days were observed with temperatures ≥30 °C for more than six hours. Relatedly, the relative humidity measured inside the canopy showed the lowest percentage during days that were ≥30 °C. The relative humidity during the daytime recorded at the weather station was constantly lower than inside the canopy. At night, the relative humidity measurements were similar. Leaf wetness within the canopy measured small increases during the night. In this measuring period, eight rain events took place, which affected the leaf wetness with increases higher than normal. Nevertheless, nine days were observed where the leaf wetness did not exceed (to support infection) 460 raw counts.

Plot II (Figure 3) also showed that during the daytime, measurements taken by the weather station showed constantly lower temperatures compared to inside the canopy. Weather station data showed only two days above 30 °C, whereas inside the canopy six days were observed with temperatures ≥30 °C for more than six hours. Relatedly, the relative humidity measured inside the canopy showed the lowest percentage during days that were ≥30 °C. The relative humidity measured by the weather station was constantly lower than inside the canopy during the daytime. During the night, the relative humidity measurements were similar. Leaf wetness within the canopy measured small increases during the night. Five rain events occurred during this measurement period, which affected the leaf wetness with increases higher than normal. Nevertheless, six days were observed where the leaf wetness did not exceed (to support infection) 460 raw counts.

As with the other plots, the measurements taken by the weather station showed constantly lower temperatures compared to inside the canopy during the daytime (Figure 4). Weather station data never exceeded 30 °C, whereas, inside the canopy, four days were observed with temperatures ≥30 °C for more than six hours. Relatedly, daytime relative humidity measured inside the canopy showed the lowest percentage during the days that were ≥30 °C. The relative humidity measured by the weather station was constantly lower than inside the canopy during the daytime. During the night, relative humidity was similar. Leaf wetness consistently increased during the night and decreased during the daytime. Extreme differences between the measured values within the canopy and the weather station occurred during the night. Measurements from the weather station showed values ≥460 raw counts every day. Measurements within the canopy showed four days during which the value did not exceed the 460 raw count threshold. In this measuring period, three rain events occurred, which affected the leaf wetness inside the canopy with increases higher than normal.

Figure 5 shows temperatures measured inside the canopy and by the ws were similar during the night. During the daytime, the canopy temperature increased faster and was up to 6 °C higher compared to the weather station. In addition, the temperatures inside the canopy decreased faster compared to the weather station. Additionally, the canopy temperatures decreased about one hour earlier compared to the temperatures measured by the weather station. Differences between the measurement heights inside the canopy appeared with temperature decreasing from 160 to 100 cm position. In summary, the weather station continuously showed lower temperatures during the daytime. Relative humidity measured both inside the canopy and by the weather station reached nearly 100% at night. Noticeable differences appeared after sunrise. Measurements within the canopy decreased faster and showed lower humidity during the day than measurements taken by the weather station. Additionally, inside the canopy, the humidity increased about one hour earlier than at the weather station. Differences between the measurement heights inside the canopy showed that the humidity decreased from 100 to 160 cm. In summary, relative humidity observed inside the canopy showed higher value changes (between day and night) than the weather station. The leaf wetness showed high differences between inside the canopy and the weather station. Measurements by the weather station showed values of ≥900 raw counts during the night and under 460 raw counts from 12 p.m. until 8–10 p.m. Inside the canopy, the leaf wetness level increased above the threshold from 12 a.m. to 12 p.m. In summary, measurements from inside the canopy showed lower changes between day and night than did the weather station. Additionally, in one case at night, the leaf wetness levels inside the canopy did not even reach the threshold.

### 3.2. Average of All Temperatures, Relative Humidity, and Leaf Wetness for Measurement Periods I–III

Average values provide an appropriate overview of the parameters measured during the experiment. To exclude aligning differences between day and night, the measurements of all parameters were separated by sunrise and sunset. Values were calculated and averaged by hour.

#### 3.2.1. Temperature

Table 2 shows that temperatures tended to increase from 100 to 160 cm inside the canopy (significant in plot I). The most noticeable differences were between the canopy average and the weather station during the daytime, with an average variation of 3.6 °C. Comparison of the weather station and canopy data showed similar night temperatures.

Figure 6 summarises data for all plots. Daytime measurements show higher variations than at night.

#### 3.2.2. Relative Humidity

Relative humidity (Table 3) decreased from 100 to 160 cm canopy heights, which were statistically significantly in nearly all plots, independent of day or night measurements. During the day, the weather station indicated on average 3.5% higher humidity than inside the canopy. During the night, higher humidity appeared inside the canopy than at the weather station.

Figure 7 summarises data for all plots. Daytime measurements show higher variations than at night.

#### 3.2.3. Leaf Wetness

Table 4 shows that leaf wetness levels increased from 100 to 160 cm. Statistically significant differences appeared during daytime and night. On average, daytime measurements were lower than night measurements. The most noticeable differences appeared between the weather station and the canopy (plot III).

Figure 8 summarises data for all plots. Night measurements had higher variations than those at daytime.

## 4. Discussion

Plasmopara viticola has two types of infection reproduction. The primary infection is the first observed infection of the season. It is caused by oospores that are overwintered inside leaf litters or vineyard debris. When conditions are favourable, oospores germinate and produce sporangium, which is transported by wind or rain and splashed on wet leaves near the ground, which may become infected. Penetration via the hyphen takes place through the stomata on the lower surface of the leaves [15]. The infection process needs a minimum of 13° C plus 8 mm of rain to occur [16]. This prerequisite enables precise forecasting, especially for the primary infection [11]. After primary infection and through the incubation time, germinated sporangia produce many zoospores, which may cause secondary infections on nearby plants. The formation of new sporangia, followed by germination and the release of new zoospores, varies from 5 to 18 days, depending on climatic conditions and varietal susceptibility of the vines [15]. Plant growth status is as equally important as the interplay of temperature, relative humidity, and leaf wetness [17]. A simple rule might be used by calculating hour–degree (hour-degree = temperature[°C] * leaf wetness duration[h]). When the critical value of 50 h–degree is reached, the conditions for possible infection are sufficient [2]. Therefore, climatic conditions have a considerable influence on the pathogen infection scenario. Secondary infection occurs more frequently during the vegetation period, which complicates the forecast. Since 1882 (when Millardet discovered the effect of copper) and the 1980s (when systemic fungicides entered the market), fungicides were applied in much greater quantities and frequencies than today [18]. In the past few years, the attempts increased to reduce these fungicide applications to minimize harmful and environmental side effects. Moreover, those applications are very costly and time-consuming and result in emissions of gas and soil compaction during application by a tractor [9]. Fungus-resistant varieties may need less plant protection than conventional varieties, but presently, there is no 100% fungus-resistant variety admitted for wine-making. With high fungal persistence, multiple fungicide applications might still be necessary. Downy mildew forecasting models help vine-growers to work in more sustainable and ecological ways by predicting infection risks and, thereby, reducing unnecessary fungicide applications [7,8,18]. Common forecasting models use simulations based on climatic weather data from field weather stations for their calculations. The models utilize recordings of temperature, relative humidity, leaf wetness, and leaf growth ratio. Additional parameters, such as rainfall, wind, and sun irradiation may supplement the simulations for more precise results and forecasting [1]. Conditions inside a vineyard can vary greatly [12] depending on the location, exposition, and the steepness of the site. That supports the claim that climatic conditions measured from a field weather station vary greatly as compared to the conditions inside a grape canopy. Presently, there is insufficient knowledge about the climatic conditions inside a grapevine canopy. In this study, significant differences were detected between data collected within the canopy and by the weather station. To exclude the influence that the placement of the sensors may have on the results of the temperatures, relative humidity, and leaf wetness, different measurement heights were used inside the canopy. Despite some differences, it turned out that the sensor height inside the canopy was less influential on the results of the measured parameters. Distinct differences were found between the measurements of the canopy average when compared to the weather station. Peña Quiñones, et al. (2020) [13] showed that day air temperature measured within the berry clusters was higher than that measured by weather stations, which is in accordance with the present study. Blaeser (1978) described the dependency of temperature compared with relative humidity in relation to the lifetime of sporangium [2]; fungus inoculum lost vitality when the temperature was above 30 °C for at least six hours. In the present study, 13 days were observed where temperatures inside the canopy increased above 30 °C for more than 6 h. In comparison, measurements taken by the weather station showed temperature conditions of ≥30 °C for more than six hours, on only two days (Figure 3). In conclusion, the weather station measured 11 fewer occurrences of conditions (causing the death of spreadable sporangia) than were measured within the canopy. In fact, inside the canopy, 9% of all experiment days showed conditions lethal to the sporangia. Whereas measurements taken by the weather station only observed 1.4% of all experiment days with lethal conditions. Therefore, using data from inside the canopy results in less predicted infection risks. Temperature and relative humidity were closely connected to the lifespan of the sporangium. Temperatures near 10 °C and relative humidity of about 70–90% led to a sporangium lifespan of 9 days. However, 10 °C and relative humidity of about 30–40% resulted in a lifespan of the sporangium of only three days. In conclusion, the higher the temperatures are, the more humidity is needed for spores to survive, but this condition also reduces spore lifespan. Temperatures of about 20 °C and relative humidity of 90% are needed for a four-day lifespan of the sporangium. Relative humidity of 70% (and 20 °C) results in only a three-day lifespan [2]. Therefore, a shorter lifespan of the sporangium lowers the exposure to a potential wet period, which is a prerequisite for infection. In this study, constantly higher temperatures in the daytime with a lower relative humidity inside the canopy were recorded, compared to the conditions recorded by the weather station. The higher temperatures and the lower relative humidity inside the canopy greatly decrease the possible lifespan of the sporangium. The differences in the recordings of those temperatures of the canopy and ws do lead to different interpretations of the sporangia lifespan and the recording of the canopy results in the prediction of a reduced successful infection compared to the weather station, which also has affects the forecast and might further reduce the fungicide application. Wet conditions permitting infections may arise independenly of a rain event; what is essential is the presence of water at the abaxial leaf surface. Depending on the level of leaf wetness, released zoospores can use the liquid media to swim with their biflagellate flagellum to reach stomata to enter the leaf surface [19]. In this study, high differences between the leaf wetness inside the canopy and by the weather station were measured. During the experiment, 16 rain events occurred, which resulted in higher leaf wetness levels than what usually appeared within the canopy. At absent rain events, the leaf wetness increased only slightly during the night. The differences between the leaf wetness inside the canopy and the weather station were significant. The leaf wetness levels measured by the weather station reached, on average, above 900 counts, which was double the leaf wetness measured within the canopy (Figure 4). The leaf wetness indicated sufficient water for infection when the raw counts were ≥460. In general, the leaf wetness measured by the weather station increased above the critical value four hours earlier in the evening than that measured within the canopy. In contrast, the leaf wetness inside the canopy remained above 460 raw counts, 1–2 h longer than at the weather station before noon (Figure 5). Due to the temperature and relative humidity measurement differences between day and night, the possibility of active sporangia, in general, is lower during the daytime. Nevertheless, the leaf wetness exceeded the critical threshold on average 2–3 h more at the weather station than inside the canopy. That said, darkness is essential for infection. Therefore, the infection possibility inside a canopy was much lower due to fewer hours of darkness with leaf wetness ≥460 raw counts. Additionally, the leaf wetness measured by the weather station showed values above the threshold (≥460 raw counts) for every day of the experiment. However, within the canopy, 19 days/nights occurred where the measured leaf wetness did not reach the threshold of ≥460 raw counts at any time. Therefore, the leaf wetness measured by the weather station indicated more infection possibilities when used in the modelling than what actually existed, as indicated by the conditions inside the canopy.

Researchers should discuss the results and how they could be interpreted from the perspective of previous studies and the working hypotheses. The findings and their implications should be discussed in the broadest context possible. Future research directions may also be highlighted.

## 5. Conclusions

Measurements taken by weather stations have historically produced the data basis for downy mildew forecasting models. However, weather stations do not accurately represent the climatic conditions inside a grape canopy where fungal disease actually occurs. This study shows that higher temperatures, lower relative humidity, and lower leaf wetness recorded inside the canopy are critically different when compared to weather station data. The discrepancies between these data have significant influences on the precision of forecast modelling. Since conditions favourable to the pathogen are not as favourable inside the canopy as those measured by the weather station, less fungicide application is likely necessary than what is currently assumed.

## Figures and Tables

**Figure 1 plants-11-01807-f001:**
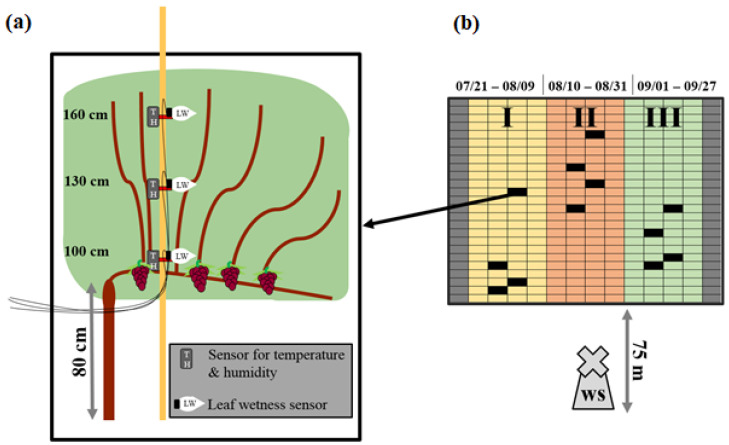
Experimental design. A canopy with different sensor heights and sensor types (**a**). Vineyard separated into three plots I–III. Each square represents one vine. Sensor-equipped vines at the plots are marked in black. Measurement periods (date) shown above (**b**). ws = weather station.

**Figure 2 plants-11-01807-f002:**
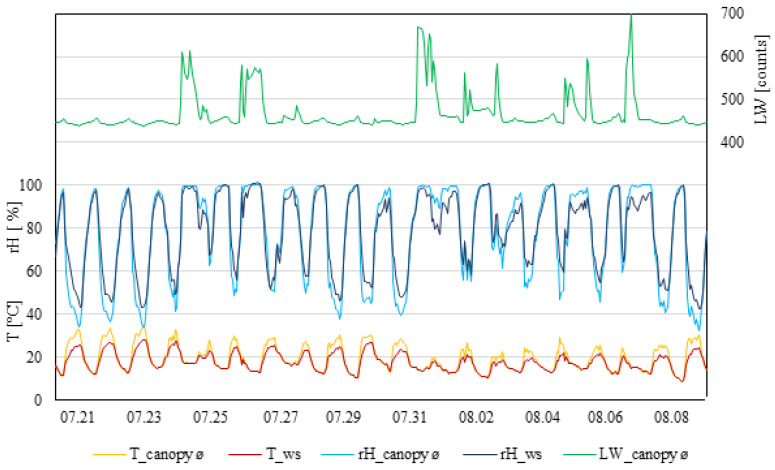
Data from measurement period I (21 July 2021–9 August 2021). T_canopy_ø, average temperature of the canopy; T_ws, temperature of the weather station (°C). rH_canopy_ø, average rel. humidity of the canopy; rH_ws, rel. humidity of the weather station (rH%); LW_canopy_ø, average leaf wetness of the canopy (counts); n = 4.

**Figure 3 plants-11-01807-f003:**
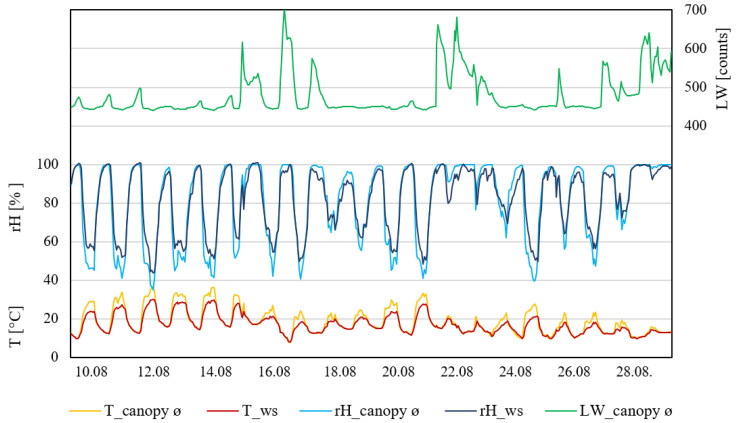
Data from measurement period II (10–31 August 2021). T_canopy_ø, average temperature of the canopy; T_ws, temperature of weather station (°C). rH_canopy_ø, average rel. humidity of the canopy; rH_ws, rel. humidity of the weather station (rH%); LW_canopy_ø, average leaf wetness of the canopy (counts); n = 4.

**Figure 4 plants-11-01807-f004:**
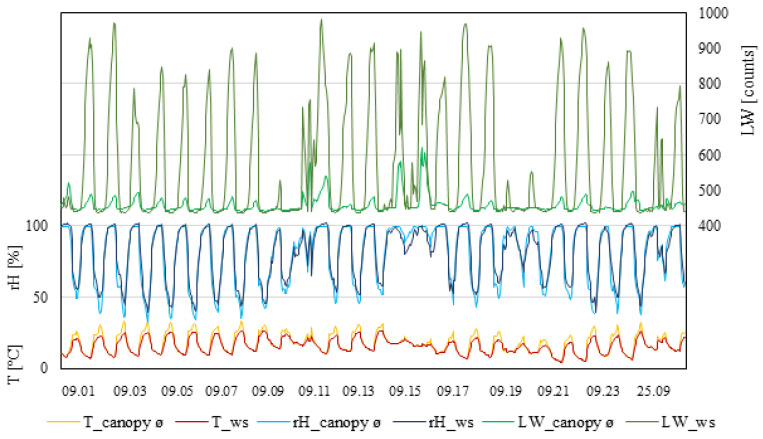
Data from measurement period III (1–27 September 2021). T_canopy_ø, average temperature of the canopy; T_ws, temperature of weather station (°C). rH_canopy_ø, average rel. humidity of the canopy; rH_ws, rel. humidity of the weather station (rH%). LW_canopy_ø, average leaf wetness of the canopy; LW_ws, leaf wetness of the weather station (counts); n = 4.

**Figure 5 plants-11-01807-f005:**
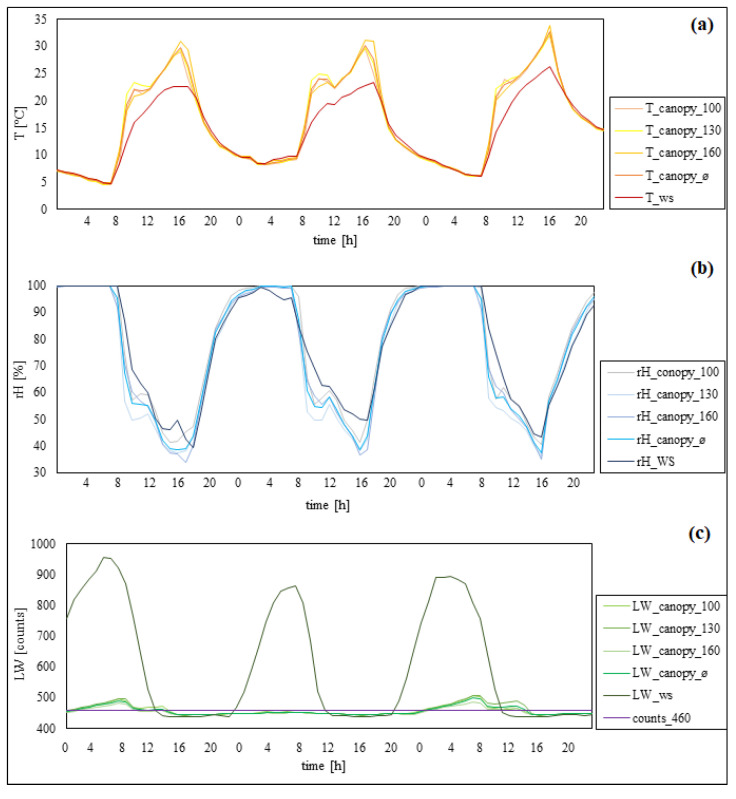
Separate illustration of temperature (**a**), rel. humidity (**b**), and leaf wetness (**c**) from 23–25 September 2021 (plot III). Weather station data, ws; different measurement heights inside the canopy, 100, 130, 160; canopy data on average, ø.

**Figure 6 plants-11-01807-f006:**
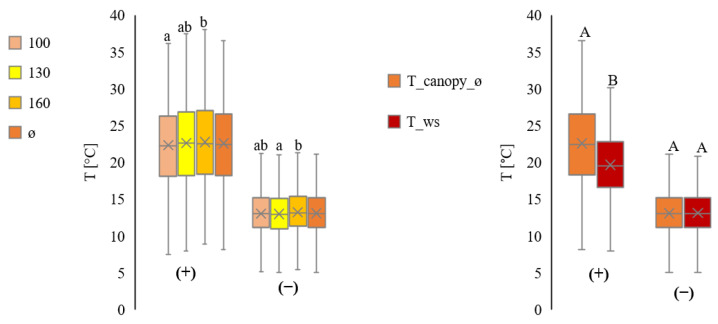
Comparison of the temperature measurements from inside the canopy at three heights (100 cm ≙ T_canopy_100, 130 cm ≙ T_canopy_130 and 160 cm ≙ T_canopy_160), and average (ø ≙ T_canopy_ø). All values in °C. Separation into day (+) and night (−) from July to September, for all plots. Comparison of the temperature measurements inside the canopy on average (T_canopy_ø) with the temperature measurements by the ws (T_ws). All values in °C. Separation into day (+) and night (−) from July to September (all plots in total). Statistically significant differences inside the canopy are indicated by lower case letters; differences between the canopy average and the weather station are indicated by capital letters, *p* < 0.05; n = 4.

**Figure 7 plants-11-01807-f007:**
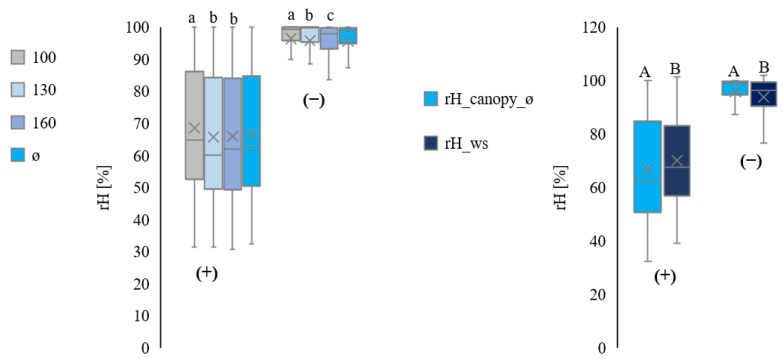
Comparison of the relative humidity from inside the canopy at three heights (100 cm ≙ rH_canopy_100, 130 cm ≙ rH_canopy_130, and 160 cm ≙ rH_canopy_160) and average (ø ≙ rH_canopy_ø). All values are in %rH. Separation into day (1) and night (2) from July to September, for all plots. Comparison of the relative humidity measurements inside the canopy on average (rH_canopy_ø) with the relative humidity measurements by the ws (rH_ws). All values are in rH% Separation into day (+) and night (−) from July to September (all plots in total). Statistically significant differences inside the canopy are indicated by lower case letters; differences between the canopy average and the weather station are indicated by capital letters, *p* < 0.05; n = 4.

**Figure 8 plants-11-01807-f008:**
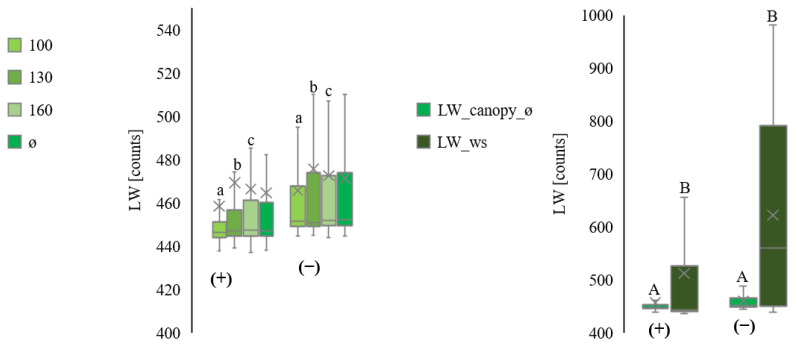
Comparison of the leaf wetness measurements from inside the canopy at three heights (100 cm ≙ LW_canopy_100, 130 cm ≙ LW_canopy_130, and 160 cm ≙ LW_canopy_160) and average (ø ≙ LW_canopy_ø). All values are in raw counts. Separation into day (+) and night (−) from July to September, for all plots. Comparison of the leaf wetness measurements inside the canopy on average (LW_canopy_ø) with the leaf wetness measurements by the ws (LW_ws). All values are in raw counts. Separation into day (+) and night (−) from September (plot III). Statistically significant differences inside the canopy are indicated by lower case letters; differences between the canopy average and the weather station are indicated by capital letters, *p* < 0.05; n = 4.

**Table 1 plants-11-01807-t001:** Fungicide application schedule. The “x” corresponds to one fungicide application. PM: powdery mildew; DM: downy mildew.

Date	15 July 2021						23 July 2021						7 August 2021						20 August 2021					
pathogen	DM			PM			DM			PM			DM			PM			DM			PM		
	Vinostar			Prosper tec			Vinostar			Luna Experience			Enervin SC &			Topas			Mildicut			Topas		
fungicide	3.6 kg/1200 L			1.2 L/1200 L			4.5 kg/1500 L			0.9 L/1500 L			Vinifol SC			360 mL/900 L			2.0 L/400 L			360 mL/400 L		
													2.7 L/900 L											
plot	I	II	III	I	II	III	I	II	III	I	II	III	I	II	III	I	II	III	I	II	III	I	II	III
application	x	x	x	x	x	x		x	x		x	x		x	x		x	x			x			x

**Table 2 plants-11-01807-t002:** Temperatures during the daytime (+) and at night (−) for each plot. Measurements inside the canopy were shown at canopy heights of 100 cm (T_canopy_100); 130 cm (T_canopy_130); and 160 cm (T_canopy_160); and 100, 130, 160 combined on average (T_canopy_ Ø). Corresponding temperatures measured by the ws (T_ws). All values are in °C. Statistically significant differences inside the canopy are indicated by lower case letters; differences between the canopy average and the weather station are indicated by capital letters, *p* < 0.05; n = 4.

(+) Plot at Daytime	T_canopy_100	T_canopy_130	T_canopy_160	T_canopy_Ø	T_ws
I	22.1 a	22.4 ab	22.8 b	22.4 A	19.8 B
II	22.2 a	22.3 a	22.6 a	22.4 A	19.7 B
III	22.1 a	19.5 a	22.3 a	21.3 A	19.5 B
(−) Plot at night	T_canopy_100	T_canopy_130	T_canopy_160	T_canopy_Ø	T_ws
I	14.3 a	14.2 a	14.4 a	14.3 A	14.3 A
II	13.8 a	13.8 a	14.2 b	13.9 A	13.8 A
III	12 a	11.9 ab	12.2 b	12 A	12.1 A

**Table 3 plants-11-01807-t003:** Relative humidity measured during daytime (+) and night (−) for each plot. Measurements inside the canopy are shown at canopy heights of 100 cm (rH_canopy_100), 130 cm (rH_canopy_130), and 160 cm (rH_canopy_160), and 100, 130, 160 combined on average (rH_canopy_Ø). Corresponding relative humidity measured by the ws (rH_ws). All values are in rH%. Statistically significant differences inside the canopy are indicated by lower case letters; differences between the canopy average and the weather station are indicated by capital letters, *p* < 0.05; n = 4.

(+) Plot.at Daytime	rH_canopy_100	rH_canopy_130	rH_canopy_160	rH_canopy_Ø	rH_ws
I	69.7 a	68.5 ab	67.4 b	68.5 A	71.2 B
II	68.9 a	67.9 a	67.2 a	68 A	71.4 B
III	66.9 a	63 b	63.8 c	64.6 A	66.3 A
(−) Plot.at night	rH_canopy_100	rH_canopy_130	rH_canopy_160	rH_canopy_Ø	rH_ws
I	95.3 a	95.4 a	94.1 b	94.9 A	92.4 A
II	97 a	96.7 a	94.7 b	96.1 A	93.8 A
III	96.9 a	95.5 b	95.1 c	95.8 A	93.4 B

**Table 4 plants-11-01807-t004:** Leaf wetness measured during the daytime (+) and night (−) separated for each plot. Measurements inside the canopy were shown at heights of 100 cm (LW_canopy_100), 130 cm (LW_canopy_130), and 160 cm (LW_canopy_160) and 100, 130, and 160 combined on average (LW_canopy_ Ø). Corresponding leaf wetness measured by the ws (LW_ws). No measures were marked with *. All values are in raw counts. Statistically significant differences inside the canopy are indicated by lower case letters; differences between the canopy average and the weather station are indicated by capital letters, *p* < 0.05; n = 4.

(+) Plot.at Daytime	LW_canopy_100	LW_canopy_130	LW_canopy_160	LW_canopy_Ø	LW_ws
I	458.3 a	467.9 b	466.6 b	464.3	*
II	466.9 a	494.6 b	483.8 c	481.8	*
III	457.4 a	459.9 b	460.8 b	459.4 A	512.5 B
(−) Plot.at night	LW_canopy_100	LW_canopy_130	LW_canopy_160	LW_canopy_Ø	LW_ws
I	464 a	468 b	471 b	467.7	*
II	472 a	511 b	491 c	491.3	*
III	460 a	460 a	460 a	460 A	622.4 B

## Data Availability

Not applicable.

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
