# Peer review of "New Aspects of In Situ Measurements for Downy Mildew Forecasting"

_plants, 2022, doi:10.3390/plants11141807_

Round 1

Reviewer 1 Report

1.     Methods: Experimental design is confusing; clarify that the successive measurement period indicates a rotation of the divided three plots. One sprayed each time and one measured.

2.      Methods: How was disease severity or healthy status defined and measured?

3.     Methods: Where were the powdery mildew fungicides used?

4.     Discussion: Are oospores solely found in the soil? Don’t we find these in leaf litter and vineyard debris?

5.     Methods: Were disease severity ratings taken throughout the study?

6.     Methods: how were the 100-160cm heights determined?

7.     Methods: should repetitions be “replicates”?

8.     Methods: Does the sentence “These infection damages would cause different micro climatic conditions inside the canopy” require citation?

9.     Methods: Measurement periods: dates for the III time periods could be mentioned in the text as well as currently shown in the figures.

10.  Abstract: applications instead of application; multiplied by instead of multiplied with

11.  Introduction: downy mildew not downEy mildew

12.  Introduction: Describe how “cold” was decided on? Describing a summer with no data and saying it was humid does not seem applicable to cold

13.  Introduction: “However, downy mildew appears not only in extremely humid years – the pathogen appears more or less every year and in every vineyard. “ This sentence is unnecessary unless re-worked to describe downy mildew throughout a season and the impact from humidity

14.  Introduction: Making the claim that grape varieties are not accepted but failing to mention that this is often due to unique genetic profiles for aroma and taste; growers and winemakers are not just avoiding the varieties for no reason.  

15.  General comment: data showing comparisons of predictions using ws versus inner canopy data would demonstrate the impact further to a reader; how significant of a change is the prediction even though we see a significance in the actual weather data?

16.  Table 1: Could mention the common or recommended spray intervals of the products

17.  Discussion: “un-necessary” can be “unnecessary”

18.  General comment: discussion demonstrates the applicability of the results in a clear and concise manner that is quickly understood.

19.  Discussion: “swim with their haustoria,” P. viticola zoospores swim with a biflagellate flagellum, not haustoria.

20.  General comment: I would address that further work is needed to confirm this study; this is still a hypothesis.  A study demonstrating a comparison between using control methods from ws and inside canopy data is necessary to move forward. We can assume these different scenarios, but field scenarios occur much differently than field and mathematical assumptions.

21.  General comment: what is the hypothesis behind the lack of leaf wetness in the canopy vs the ws? Why wouldn’t the canopy appear to hold excess moisture with a lack of air flow?

22.  Could include plot titles for clarity

Author Response

Dear Sir or Madame,

Thank you very much for reading and commenting my article.

I tried to implement as much as possible. For some points, I have some comments or questions back to you.

  1. Yes, I monitored the disease severity of all the plots during their specific measurement period. I used that for the switch from one to the next plot and will use it in a following study.
  2. The different measuring heights were determined from the ground. I tried to show it in figure 1.
  3. I´m not sure if this needs citation. My thought was; When leafs get necrotic and roll they induce different temperature distribution inside the canopy and cannot transpire anymore or even worse, they of fell of the vine. Then the supposition is close that the inter canopy climate will change compared to a healthy vine. Do you think I should explain this a bit better in the article?
  4. I thought I explained the importance of humidity and leaf wetness and temperature a few sentence before. If you think this is to less, I think I can delete the sentence then.
  5. Exactly this, compared with the disease monitoring will be in the next following article of me.
  6. Thanks and yes, you are right. As already mentioned a following study is in progress.
  7. Leafs from the upper part of the canopy were protecting leafs from the lower part of the canopy. Dew or rain will be in first contact with the upper leafs and not with the whole canopy at once. The sensor at a weather station only represents one leaf, at the upper part. However, downy mildew can appear in all parts of the canopy.

Thank you really, really much for your helpful comments.

Kind regards,

Melissa Kleb

Reviewer 2 Report

Despite of significant difference among the environment data, I believe that these difference does not interfere  at the disease epidemiology in all sense.

Author Response

Dear Sir or Madame,

thank you for reading my article.

Suitable for your suggestion, I am working on a following study that will show the differences calculated by a forecasting model compared with monitoring. So I can make sure if it has any real impact or not. If you are interested in these results let me know.

Thank you really much!

Kind regards,

Melissa Kleb

Reviewer 3 Report

The manuscript “New Aspects of in-situ Measurements for Downy Mildew Forecasting” studied the conditions of infection potential in viticulture of Plasmopara viticola. The manuscript described well the infection process, but it isn’t presented a method of infection forecasting as title suggests. All results presented confirm the knowledge from literature but didn’t add something new. It should develop a method of infection based on the main bioclimate indexes with computing the minimum favorable conditions for infection and incubation time able to forecast the infection.

Moreover, the experiment design does not have sense. Why in the three plots the data was collected in different points? Why in the three plots the data was collected in different periods (plot I from 21.07-08.08 and Plot II+II from 01.09 to 25.09)? This data differs very much so, cannot be compared.

In this form, the manuscript could not be suitable for publication in Plants journal, because there are major shortcomings of the method and nothing new is added.

Author Response

Dear Sir or Madame,

thanks for reading my article.

Maybe there is some unclarity, which I would like to explain. 

The data was collected in different points to guarantee randomised data which can be seen as average of the belonging plot. The measurement devices were switched from one plot to another. During the time when the sensors were placed inside a plot, this plot was not protected by fungicides against downy mildew, so we had to switch to the next uninfected plot after a while. If we had not switched the devices to the next plot, the vines could not represent a usual canopy climate. After long infection with no protection the leafs get necrotic or in worst case, the vine looses all the leafs and would not provide the climatic conditions. Healthy or mostly healthy leafs generate different temperature distribution inside the canopy, they can transpire and were able to collect water at the abaxial leaf surface, no leafs or highly negrotic leafs cannot provide this. The comparison of the plots was not direct. Much more the data of the plots collected inside the canopy was compared to the weather station. Also plot II and III were not at the same time, figure 1 shows the measuring period for each plot.

I tried to clarify these aspects a bit more in the actual version of this article. Thanks for your comments.

Kind regards,

Melissa Kleb

Round 2

Reviewer 3 Report

Dear Authors,

I appreciate your effort to clarify the paper but in the new version of manuscript “New Aspects of in-situ Measurements for Downy Mildew Forecasting” there were includes a few text changes, but the paper should be re-organized and expanded to include a new approach that brings important results in the field. Improvement / modification of measurement techniques cannot be considered an original and an important topic for the scientific community, unless it produces results with major impact in control of the pathogens.

The influence of environmental conditions, such as temperature and relative humidity, and vegetative state (leaf wetness) on infection with downy mildew are already done, even the forecasting model used in this paper. Moreover, the day-night difference between environmental conditions in open field (weather station) and inside a grape canopy are known. The presented results confirm the theory used so far, without having a major original component.

Consider above observation, I maintain my first opinion that the paper can not be published in PLANTS journal in this form.